# Optomechanical Microwave-to-Optical Photon Transducer Chips: Empowering the Quantum Internet Revolution

**DOI:** 10.3390/mi15040485

**Published:** 2024-03-31

**Authors:** Xinyao Xu, Yifei Zhang, Jindao Tang, Peiqin Chen, Liping Zeng, Ziwei Xia, Wenbo Xing, Qiang Zhou, You Wang, Haizhi Song, Guangcan Guo, Guangwei Deng

**Affiliations:** 1Institute of Fundamental and Frontier Sciences, University of Electronic Science and Technology of China, Chengdu 610054, China; 2021190502001@std.uestc.edu.cn (X.X.); 2021190502021@std.uestc.edu.cn (Y.Z.);; 2Key Laboratory of Quantum Physics and Photonic Quantum Information, Ministry of Education, University of Electronic Science and Technology of China, Chengdu 611731, China; 3Southwest Institute of Technical Physics, Chengdu 610054, China; 4CAS Key Laboratory of Quantum Information, University of Science and Technology of China, Hefei 230026, China; 5Institute of Electronics and Information Industry Technology of Kash, Kash 844000, China

**Keywords:** optomechanics, quantum transducer, quantum Internet, quantum chip

## Abstract

The first quantum revolution has brought us the classical Internet and information technology. Today, as technology advances rapidly, the second quantum revolution quietly arrives, with a crucial moment for quantum technology to establish large-scale quantum networks. However, solid-state quantum bits (such as superconducting and semiconductor qubits) typically operate in the microwave frequency range, making it challenging to transmit signals over long distances. Therefore, there is an urgent need to develop quantum transducer chips capable of converting microwaves into optical photons in the communication band, since the thermal noise of optical photons at room temperature is negligible, rendering them an ideal information carrier for large-scale spatial communication. Such devices are important for connecting different physical platforms and efficiently transmitting quantum information. This paper focuses on the fast-developing field of optomechanical quantum transducers, which has flourished over the past decade, yielding numerous advanced achievements. We categorize transducers based on various mechanical resonators and discuss their principles of operation and their achievements. Based on existing research on optomechanical transducers, we compare the parameters of several mechanical resonators and analyze their advantages and limitations, as well as provide prospects for the future development of quantum transducers.

## 1. Introduction

Grounded in quantum mechanics yet adhering to classical physical laws, computer technology and the Internet arguably rank among the most significant scientific and technological advancements of the 20th century. As we enter the 21st century, quantum information emerges at the intersection of quantum mechanics and information science, propelling the development of information science into a new “quantum” era [1]. With breakthroughs continually achieved by researchers worldwide, “quantum information” is gradually transitioning from the laboratory to application. However, to realize a true “quantum Internet”, “quantum interconnects” is a difficult hurdle that must be overcome.

Similar to the classical Internet (Figure 1 shows the comparison of classical and quantum computer networks), quantum interconnects involve linking different systems (or platforms) that perform quantum information processing and operations, thereby allowing the transfer of quantum states between different physical parts or degrees of freedom within a system [2,3]. Through this transfer, quantum subsystems operating at different energy scales can be integrated, enabling efficient and scalable distributed quantum networks.

However, due to losses, lifetime limitations, etc., it is difficult for existing technologies to achieve long-range transmission of solid-state quantum bits, which poses many challenges for our quantum interconnections. Microwave–optical quantum transducer may be a good solution to this problem. Although quantum information processing typically operates at microwave frequencies, the most mature long-distance information transmission systems currently available are optical fibers, and the loss of transmitting photons through optical fibers is much lower in coaxial cables [4]. Therefore, developing transducer chips that efficiently couple microwave photons with telecom photons is critical. Many possible transducer schemes have been proposed and achieved promising results, utilizing platforms such as optomechanics [5,6,7,8], electro-optics [9,10,11,12], optomagnonics [13,14,15,16,17], atomic ensembles [18,19,20,21], etc. [22,23,24].

Among all the current transducer solutions, optomechanical systems maintain the highest efficiency in conversion rates. Due to the ability of a mechanical oscillator to couple equivalently to fields of different wavelengths, it is well placed to act as an intermediary to bridge the gap of five-orders-of-magnitude between the gigahertz microwave and the telecom optical photons [22]. Simultaneously, the controllability of optical cavities endows optomechanical systems with high tunability and flexibility. They are also relatively easy to integrate on-chip. The key feature of optomechanical quantum transducers, as shown in Figure 2, is the utilization of mechanical resonators as intermediaries to complete the microwave-to-optical conversion via microwave–phonon and phonon–photon. Available mechanical resonators include membrane films [25,26,27,28], phononic crystals [29,30,31,32,33,34,35,36], microdisks [37,38], and bulk acoustic wave resonators [39,40], forming a diverse array of optomechanical transducer structures with different performances.

Our review focuses on promising on-chip optomechanical quantum transducers. We classify these transducers into four categories based on mechanical resonator type. The advantages and disadvantages of different types of transducer systems composed of these mechanical resonators are discussed. Additionally, we analyze current bottlenecks and challenges encountered in these systems and propose possible solutions. Finally, we compare the parameters of the four types and discuss the future development of transducer chips.

## 2. Optomechanical Transducer Systems

The discipline dedicated to the preparation, measurement, and manipulation of quantum states of mechanical (or acoustic) modes within cavity optomechanical systems is known as quantum cavity optomechanics [41]. Its primary focus lies in the interaction between photons and phonons. In recent years, researchers have been drawn to the study of phonons due to their advantages such as localizability, capability of phonon–phonon interaction, and the ability to couple with various quantum systems including optical fields, electric fields, microwave fields, diamond NV centers, and more [42]. Mechanical resonators, serving as carriers of phonons, can efficiently couple microwave circuits and photonic resonators, thereby acting as a medium to achieve bidirectional conversion between microwave and optical photons. With the continuous development of cavity optomechanics, hybrid systems combining microwave circuits, mechanical and photonic resonators to couple electricity acoustics, and light have demonstrated unique advantages in the field of quantum transducers, achieving a remarkable array of results [22,41,43]. The following will introduce some optomechanical principles and discuss transducer systems mediated by different mechanical resonators: membrane resonator, phononic crystal resonator, microdisk, and bulk acoustic wave resonator.

For quantum transducers, the most important and fundamental principles of optomechanics are radiation pressure and optomechanical coupling. Radiation pressure, which involves the momentum transfer of photons, fundamentally links the behavior of the optical radiation field within a cavity to its mechanical dynamics. The Hamiltonian describing a coupled system of an optical cavity and microwave resonator resonating at $\omega_{o}$ and $\omega_{e}$, respectively, interacting with a mechanical resonator oscillating at $\omega_{m}$, is [4,43,44,45]
(1)$$\hat{H}=\hbar \omega_{m}{\hat{b}}_{m}^{†}{\hat{b}}_{m}+\hbar \underset{j=o,e}{\sum }[\omega_{j}{\hat{b}}_{j}^{†}{\hat{b}}_{j}+{\hat{H}}_{int}+{\hat{H}}_{d}]$$
where ${\hat{b}}_{j}$ (j=o,e) and ${\hat{b}}_{m}$ refer to optical/microwave and mechanical annihilation operators, respectively. The initial pair of terms represent the independent Hamiltonians for both the mechanical motion and the cavity optical field, while ${\hat{H}}_{int}=-\hbar g_{0,j}\left({\hat{b}}_{m}+{\hat{b}}_{m}^{†}\right){\hat{b}}_{j}^{†}b_{j}$ describes the dispersive coupling intrinsic to the optomechanical system, with the single-photon optomechanical (electromechanical) coupling strength $g_{0,j}=g_{om}$ ($g_{0,j}=g_{em})$. When the cavity is driven by a coherent electromagnetic field of laser frequency ω and amplitude ε, ${\hat{H}}_{d}=\hbar \varepsilon ({\hat{b}}_{j}^{†}e^{-i\omega t}+$ ${\hat{b}}_{j}e^{i\omega t})$.

A frame rotating with the drive is chosen, and the laser drive $\Delta_{j}=\omega -\omega_{j}$ is introduced with $\Delta_{j}<0\;(\Delta_{j}>0)$ termed red (blue) detuning. The linearized Hamiltonian then becomes
(2)$${\hat{H}}_{lin}=\hbar \omega_{m}{\hat{a}}_{m}^{†}{\hat{a}}_{m}+\hbar \underset{j=o,e}{\sum }\left[-\Delta_{j}{\hat{a}}_{j}^{†}{\hat{a}}_{j}+g_{j}\left({\hat{a}}_{j}e^{i\Delta_{j}t}+{\hat{a}}_{j}^{†}e^{-i\Delta_{j}t}\right)\left({\hat{a}}_{m}e^{-i\omega_{m}t}+{\hat{a}}_{m}^{†}e^{i\omega_{m}t}\right)\right]$$
where ${\hat{a}}_{j}={\hat{b}}_{j}-\sqrt{{\bar{n}}_{cav,j}}$ and $g_{j}=g_{0,j}\sqrt{{\bar{n}}_{cav,j}}$ is the optomechanical (electromechanical) coupling strength. ${\bar{n}}_{cav,j}=4{\left|\varepsilon \right|}^{2}/\left(\kappa_{j}^{2}+4\Delta_{j}^{2}\right)$ refers to the mean number of photons in the cavity and its amplitude can amplify the optomechanical coupling strength, where $\kappa_{j}$ is the total loss rate of optical (microwave) mode.

Adjusting the detuning enables diverse interactions between the optical (microwave) and mechanical systems. By setting the red detuning of $\Delta_{j}=-\omega_{m}$ and applying the rotating-wave approximation to eliminate terms oscillating at $\pm 2\omega_{m}$, the Hamiltonian (2) in a frame with respect to $-\hbar \Delta {\hat{a}}_{o}^{†}{\hat{a}}_{o}+\hbar \omega_{m}{\hat{a}}_{m}^{†}{\hat{a}}_{m}$ simplifies to
(3)$${\hat{H}}_{BS}=\hbar \underset{j=o,e}{\sum }g_{j}\left({\hat{a}}_{j}{\hat{a}}_{m}^{†}+{\hat{a}}_{j}^{†}{\hat{a}}_{m}\right)$$
where photons and phonons undergo excitation exchange at a rate g, denoted as “beam-splitter” interaction [43]. Transducers are often designed with red detuned driving to facilitate sideband cooling, where scattered photons interact with mechanical modes of higher frequency, thereby extracting phonons’ energy and cooling them to ground state.

By setting the blue detuning of $\Delta_{j}=\omega_{m}$ and eliminating terms oscillating at $\pm 2\omega_{m}$, the Hamiltonian becomes a parametric down-conversion
(4)$${\hat{H}}_{PDC}=\hbar \underset{j=o,e}{\sum }g_{j}\left({\hat{a}}_{j}{\hat{a}}_{m}+{\hat{a}}_{j}^{†}{\hat{a}}_{m}^{†}\right)$$
which can generate entanglement between optical (microwave) photons and phonons [46].

Figures of merit of a transducer, including signal transfer efficiency (η), bandwidth (BW), added noise ($N_{add}$), and quality factor (Q factor), need to be measured to evaluate the transducer performance of quantum information conversion [47].

The signal transfer efficiency (η) is the gain in terms of optical sideband photons to the microwave quanta extracted from the source [47,48]. When the parametric pump is red-detuned by the resonant frequency of the intermediate mechanical mode, the internal (without considering photon extraction ratios ($\eta_{o}$ and $\eta_{e}$) at external ports) photon number conversion efficiency ($\eta_{int}$) within transducer at zero detuning between microwave and optical modes is [22]
(5)$$\eta_{int}=\frac{\frac{4\kappa_{o}g_{om}g_{em}}{\kappa_{e}}}{{\left(\gamma +\kappa_{o}+\frac{g_{om}g_{em}}{\kappa_{e}}\left[1-\frac{{\left(\kappa_{e}/2\right)}^{2}}{{\left(2\omega_{m}\right)}^{2}+{\left(\kappa_{e}/2\right)}^{2}}\right]\right)}^{2}}$$

Introducing the cooperativities $C_{jm}=4g_{jm}^{2}/\kappa_{j}\gamma$, (j=o,e) between j and mechanical modes with the damping rate of the mechanical resonator γ, the internal efficiency becomes
(6)$$\eta_{int}=\frac{4C_{em}C_{om}}{{\left(1+C_{em}+C_{om}\right)}^{2}}$$

Then, the total microwave-to-optical photon conversion efficiency is $\eta =\eta_{o}\eta_{e}\eta_{int}$, where the photon extraction ratios are the fractions of external coupling rates ($\kappa_{o,ext}$ and $\kappa_{e,ext}$) and the total loss rates, i.e., $\eta_{o}=\kappa_{o,ext}/\kappa_{o}$ and $\eta_{e}=\kappa_{e,ext}/\kappa_{e}$. This expression equation shows that to obtain higher microwave–optical conversion efficiency (η→1), it is necessary to meet conditions of large cooperativities $C_{em}=C_{om}\gg 1$, and highly over-coupled ports ($\eta_{o}$ and $\eta_{e}\to 1$).

The transduction bandwidth (BW) is the band of frequencies over which a high-fidelity conversion is achieved
(7)$$BW=\gamma +\kappa_{o}+\frac{g_{om}g_{em}}{\kappa_{e}}\left[1-\frac{{\left(\kappa_{e}/2\right)}^{2}}{{\left(2\omega_{m}\right)}^{2}+{\left(\kappa_{e}/2\right)}^{2}}\right]$$

A broader instantaneous bandwidth is essential for aligning the temporal mode of photonic quantum states in high-fidelity deterministic quantum network protocols [49]. Equation (6) shows that the bandwidth could be increased significantly by stronger external and intermode coupling rates while keeping a low loss rate of microwave, mechanical, and optical cavity modes.

Apart from high conversion efficiency and broad bandwidth, maintaining a low added noise ($N_{add}$) is also important; this is quantified as the equivalent input number of added noise quanta, i.e., the output noise scaled by the inverse of the conversion efficiency [43]. $N_{add}$ denotes the minimum number of photons required per mode in an input signal to surpass the noise threshold. In particular, $N_{add}\leq 1$ means sensitivity to single photons, which is desirable for efficient microwave–optical phonon conversion. The added noise mainly consists of environmental thermal noise and undesired parametric processes. In most transducers discussed in this paper, the latter is mitigated by a factor of ${\left(\frac{\kappa_{j}}{\omega_{j}}\right)}^{2},$ (i=o,e) in the deep resolved-sideband regime, and is relatively small compared with environmental thermal noise. Assuming that inputs from various environmental noise sources are uncorrelated, the environmental noise can be described by its average thermal occupation $n_{th}=\frac{1}{\;\left(e^{\hbar \omega /k_{B}T}-1\right)}$. Considering the large optical resonance frequency, we can neglect the environmental noise $n_{th,o}$ of optics, even at room temperature, and only consider that from microwave (${\bar{n}}_{th,e}$) and mechanical (${\bar{n}}_{th,m}$) thermal baths. We then have $N_{add}=N_{add,e}+N_{add,m}$, which are
(8)$$N_{add,e\left(up\right)}=\left(\frac{1}{\eta_{e}}-1\right){\bar{n}}_{th,e}$$
(9)$$N_{add,m\left(up\right)}=\left(\frac{1}{\eta_{e}C_{em}}-1\right){\bar{n}}_{th,m}$$
for microwave-to-optical phonon conversion (up-conversion), and
(10)$$N_{add,e\left(down\right)}=\frac{1-\eta_{e}}{\eta_{o}}\frac{{\left(1+C_{om}\right)}^{2}}{C_{em}C_{om}}{\bar{n}}_{th,e}$$
(11)$$N_{add,m\left(down\right)}=\frac{1}{\eta_{o}C_{om}}{\bar{n}}_{th,m}$$
for optical-to-microwave phonon conversion (down-conversion). It is also seen that over-coupling at the microwave port and large cooperativities can reduce the overall added noise.

Importantly, the quality factor (Q factor) serves as a crucial parameter for evaluating the performance of a transducer. Its expressions are $Q_{i}=\frac{\omega_{i}}{\kappa_{i}}$, (i=e,o), for microwave/optical mode with a linewidth of $\kappa_{i}/2\pi$, and $Q=\frac{\omega_{m}}{\gamma }$ for mechanical mode with a linewidth of γ/2π. Since lower energy dissipation in the system signifies enhanced precision and stability, and higher resonant frequencies result in reduced environmental thermal noise, both of which contribute to a larger Q factor, transducers should strive for a higher Q factor to achieve superior conversion performance. Nevertheless, a high Q factor may come with a trade-off of low transduction bandwidths, since a small linewidth (low loss rate) can lead to a narrow bandwidth, as discussed before.

The following sections will discuss transducers based on four types of mechanical resonator platforms: membrane, photonic crystal, microdisk, and bulk acoustic wave resonators.

### 2.1. Membrane Resonator

The use of membranes as mechanical resonators to accomplish microwave-to-optical photon conversion was the first to be implemented and is currently the most efficient conversion scheme of all optomechanical transducers [25,26,50]. This membrane resonator structure originated from the optimization of Fabry–Perot cavities, where the optical field circulating in the cavity acts on the moving mirror, causing it to displace and in turn interact with the optical cavity mode [51]. When the membrane is placed between two fixed optical mirrors, the oscillation of the membrane replaces the movement of the mirror, thereby segregating the optical and mechanical functionality into different physical structures. Since the thickness of the membrane is very small, its overlap with the optical field is minimal, resulting in negligible optical losses. Furthermore, passive laser cooling on the membrane is 100 times greater than in previous mechanical devices, allowing us to have a cavity with high Q factor [52,53].

The key to realizing transduction in this membrane resonator lies in its simultaneous coupling with both light and microwaves. As illustrated in Figure 3, a portion of the membrane is coated with conductive materials (such as niobium) [26], which is located within the LC circuit and also serves as part of the capacitor. Another portion of the membrane is situated within the Fabry–Perot cavity. When the membrane vibrates, the microwave and optical resonance frequencies are modulated simultaneously, completing the information exchange [54,55,56].

The earliest research using this structure was conducted in 2014 by Bagci et al. [48], who designed a strong coupling electro-optomechanical resonator using SiN film. Their study laid the groundwork for coherent microwave-to-optical conversion.

Subsequently, Andrews et al. [26] achieved the first true bidirectional microwave–optical conversion, with optical conversion efficiency ($\eta_{o}$) of 11% and a conversion efficiency of ~10% for classical signals [26]. However, in their experiment, vibrations driven by thermal fluctuations introduced approximately 1700 quanta of noise, significantly limiting the conversion efficiency [3]. Another article in 2018 proposed effective solutions with sideband cooling and feedforward protocol to achieve a higher total conversion efficiency of 47% (with $\eta_{o}$ ∼ 0.5, $\eta_{e}$ ∼ 0.9) [54], while the noise level was only 38 photons. This performance is highly impressive, although it still falls short of the 50% threshold for quantum state transfer and the sub-photon noise level required for quantum operations [3,22,26,56].

It is worth mentioning that this structure has successfully achieved the complete conversion from quantum bits to optical photons, which is also one of the ultimate goals in designing quantum transducers [2,25]. In this experiment, the researchers designed a very delicate structure to achieve isolation: by integrating the circuit quantum electrodynamics (cQED) module responsible for reading out superconducting quantum bits and the electro-optomechanical transducer responsible for conversion onto two separate chips, supplemented with circulators and directional coupler, the interaction between the transducer’s pumping and the quantum bits was minimized. What is even more exciting is that the transducer operates continuously rather than in pulses, resulting in an impressive maximum transmission efficiency of $1.9\times 10^{-3}$, which surpasses previous studies by more than two orders of magnitude [25].

In addition to SiN, NbTiN and 2D materials such as hexagonal boron nitride (h-BN) and graphene are also potential choices for membrane materials [27,50,57]. Transducers utilizing NbTiN membranes achieve a conversion efficiency of 47% while reducing additional noise to only 3.2 photons [50]. The mechanical modes of 2D resonators have also been shown to possess excellent electrostatic tunability, facilitating strong coupling between microwaves and mechanics [27,57].

The primary advantage of quantum transducers utilizing membrane resonators is their exceptionally high conversion efficiency. Different from integrated photonic resonators, usually with nanofabrication flaws, Fabry–Perot has the advantage of a higher Q factor and reduced dissipation rates [22]. Its large mode volume also enables the cavity to hold large amounts of pump photons to elevate the optomechanical coupling rate to tens of kHz [22]. Moreover, the microwave–mechanical coupling achieved through electromagnetic effects also boasts a high electromechanical conversion efficiency. Therefore, membrane resonators stand out as the most efficient transducer design among all optomechanical principles utilized to date, with internal conversion efficiencies as high as ~99% in good construction [54]. The strong synergy between optomechanical and electromechanical interactions enables efficient bidirectional conversion to become a reality.

However, the most significant challenge in further optimizing this structure might be cooling. When operating at low temperatures, the requirement for a high-power pump during the implementation of optomechanical coupling results in the generation of substantial heat, making sideband cooling challenging to achieve. Passive cooling methods, on the other hand, impose significant limitations on the membrane structure [22,50,58,59]. Additionally, 3D Fabry–Perot cavities often rely on manual construction, which are not conducive to industrial production or large-scale on-chip integration [22]. The final drawback lies in the low bandwidth supported by this structure; the bandwidths of the two experiments with optimal conversion rates were both at the kHz level [50,54].

### 2.2. Phononic Crystal Resonator

Phononic crystals are currently among the most favored mechanical resonators by researchers and are composed of periodically arranged atoms or molecules. Similar to crystals, where the potential fields formed by the periodic arrangement of atoms give rise to electronic bands and band gaps, phononic crystals also possess phononic bands and band gaps. There are periodic regions (within a phononic band gap for a desired breathing mode frequency) on either side of a phononic crystal, which suppresses phonon propagation, and thus confines phonons to propagate only within manually set structural defects (in the phonon band frequency ranges) in the center of the phononic crystal. These phonons are confined to narrow phononic Brillouin zones, where they can strongly couple with photons also localized and confined within the same area [60].

Microwave–mechanical coupling using phononic crystal resonators can be achieved in two different ways: through the piezoelectric or the electromagnetic effect. For transducers utilizing the piezoelectric effect, one common operational mode involves utilizing interdigital transducers (IDTs) to convert the electrical signals transmitted through microwave waveguides into surface acoustic wave (SAW) signals. Subsequently, these SAW signals are guided into the phononic crystal through a phononic waveguide, operating in a “two-stage mode”, as shown in Figure 4a. For phononic crystal transducers utilizing the electromagnetic effect (Figure 4b), their operating principle is similar to membrane transducers. They utilize the phononic crystal as a mechanical resonator to simultaneously modulate the coupling between microwaves and light [31,61].

The optomechanical coupling process typically occurs within the structure of integrated optomechanical crystals (OMCs), which utilize the simultaneous formation of photonic and phononic energy bands and bandgaps through the same periodic structure [62]. Through the design of the periodic and defect regions of the OMC structure, a high mechanical frequency (~GHz) (see in Table 1), communication optical band (~1550 nm), and strong optomechanical coupling rate ($g_{om}∕2\pi \;\sim$ MHz) can be achieved [60,62,63].

As early as 2013, Bochmann et al. [64] demonstrated coherent interactions between microwave, mechanical, and optical modes using such a piezoelectric OMC, suggesting its potential for realizing microwave-to-optical quantum state transfer. Their OMC achieved optical and mechanical Q factors of 13,000 and 2500, respectively. Although conversion efficiency was not discussed, as well as some other important parameters of transducers, their work was pioneering.

In 2016, Balram et al. [65] utilized such an OMC cavity piezo-optomechanical transducer to design a transduction platform using GaAs, characterized by two IDTs “sandwiching” an optomechanical crystal in the middle. The phononic waveguide, created by removing a row of cross-shaped defect geometries, serves to minimize dissipation of surface acoustic waves during transmission. Although the electrical signals used in this experiment were RF instead of base state microwave signals, and the overall conversion efficiency was not high ($\sim 1.5\times 10^{-13}$), it represented a significant advancement in the field.

Material platforms used for the OMC cavity transducer include AlN [36,62], membrane lithium niobate (LN) [33,34,35,66], silicon-on-insulator (SOI) [31,61], gallium arsenide (GaAs) [36,65], and gallium phosphide (GaP) [67,68]. Bidirectional microwave-to-optical photon conversion at room temperature (T ~300 K) has been observed in an AlN piezo-optomechanical crystal with ends coupled by Lamb wave interdigital transducers (IDTs) [69]. As shown in Figure 4c, its total and internal conversion efficiency is $\eta \sim 10^{-8}$ and $\eta_{int}\;\sim \;6.5\times 10^{-3}$, respectively [69]. After that, a higher conversion efficiency of $\eta \;\sim \;10^{-5}$ at room temperature has been achieved in a 1.8 GHz LN transducer (Figure 4e) with increased $\eta_{int}\;\sim \;2.6\times 10^{-2}$ and optomechanical cooperativity $C_{om}\;\sim \;1.2\times 10^{-5}$ [34]. This shows that second-order nonlinearity materials such as LN exhibit low optical losses to sustain high Q-factors in OMCs, and a higher piezoelectric coupling rate for increased energy involvement within the optomechanical cavity. Recently, the conversion efficiency increased to 0.09 in a Si-LN-MoRe piezo-optomechanical transducer, where the OMC cavity is linked to a ${\mathrm{LiNbO}}_{3}$ piezoelectric block by a phononic waveguide [33,70]. Improvements can be made in lower intrinsic optical linewidths of SOI optomechanical photonic crystal cavities, and the optical resonator linewidth through fabrication optimization of the LN block to further reduce its noise ($N_{add}\;\sim \;7$) [58,63]. Another 1D nanobeam OMC on the LN platform has achieved $C_{om}>1\;$and induced phonon lasing (Figure 4d) [66]. This phenomenon, which happens when the optomechanical backaction surpasses the intrinsic mechanical dissipation of the system, is essential for efficient phonon–photon conversion [71]. Such a system demonstrates optical Q factors exceeding $3\times 10^{5}$ in a communication optical band λ ≈ 1550 nm, and a mechanical mode frequency ~ 2 GHz with Q factors approximately $1.4\times 10^{4}$ at 4 K. Optomechanical cooperativity $C_{om}$ larger than 1 was also achieved in a GaAs OMC piezo-optomechanical transducer ($C_{om}\;\sim \;1.73$) [36]. The increased optomechanical cooperativity and stronger photoelastic effect of GaAs gives larger single-photon optomechanical coupling ($g_{om}/2\pi \sim$ 1 MHz), though its piezoelectric effect is weaker than AlN and LN [36].

**Figure 4 micromachines-15-00485-f004:**
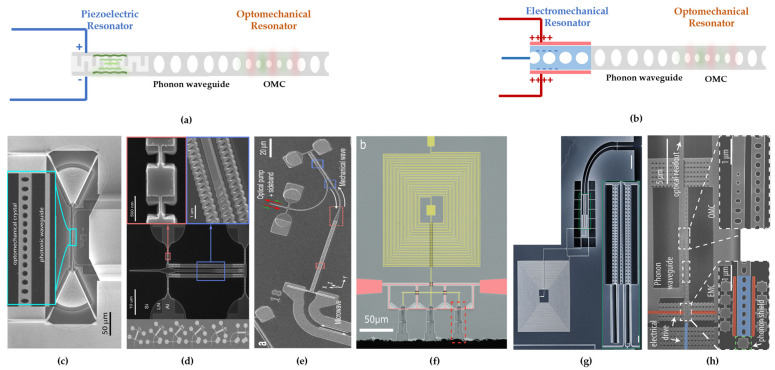
OMC cavity piezo-optomechanical (**a**,**c**–**f**) and electro-optomechanical (**b**,**g,h**) photon transducers. (**a**) Schematic diagram of phononic crystal simultaneously coupling with piezoelectric mechanical resonator and photonic crystal. (**b**) Schematic diagram of phononic crystal simultaneously coupling with electromechanical resonator and photonic crystal. (**c**) Room temperature OMC–Lamb wave IDTs transducer [69]. (**d**) OMC cavity piezo-optomechanical microwave-to-optical phonon transducer based on LN [66]. (**e**) Room temperature OMC cavity transducer based on LN [34]. (**f**) GaP microwave-to-optics transducer with low noise [67]. (**g**) Integrated OMC cavity electro-optomechanical photon transducer [61]. (**h**) Electro-optic transduction by OMC cavity transducer on SOI [31]. (**c**) is reprinted from Vainsencher et al., Appl. Phys. Lett. 109. 033107 (2016) [69] with the permission of AIP Publishing. (**d**,**h**) are reproduced from [31,66] under the terms of the Optica Open Access Publishing Agreement. (**e**–**g**) are reproduced from [34,61,67] under Creative Commons Attribution 4.0 International License.

One of the most successful experiments in this structure was conducted by Painter’s group [29,30], which achieved the complete process from excitation of superconducting qubits to optical photon transmission. They employed a hybrid platform of AlN-on-Si, designing the dimensions of the piezoelectric cavity to match both the periodic mechanical modes and the periodicity of the IDTs, thereby significantly enhancing the electromechanical coupling rate. The single-phonon initialization probability reached 75%. The experiment was carried out at low temperatures (15 mK) and under low power pumping ($n_{cav,o}=44$), successfully reducing the added noise to a sub-photon level ($N_{add\;}\sim \;0.57$) and achieving a total conversion efficiency of $\sim \;10^{-5}$ [29,30].

Furthermore, the group’s recently published paper utilized the transducer structure to prepare and detect optical–microwave photon pairs [72]. Observing phenomena such as anti-bunching confirms the nonclassical correlations between these photon pairs, which is crucial for achieving distributed entanglement in quantum networks. Another cutting-edge work from their group has achieved even further progress [73]. By using spontaneous parametric down-conversion to generate microwave–optical photon pairs and demonstrating their entanglement through time-phase encoding, this entanglement can facilitate direct connections between quantum communication and computing platforms, marking another significant breakthrough in the field of converting solid-state qubits to communication photons.

An integrated cavity electro-optomechanical photon transducer developed by a traditional SOI platform is more scalable, integrable, and compatible with CMOS technology than 3D cavities [61]. It features a photonic crystal cavity with aluminum-coated nanostrings linked to a superconducting LC resonator, as shown in Figure 4g, and achieved a high optomechanical coupling rate of $g_{om}∕2\pi ∼0.66\;$MHz [61]. The lack of reliance on piezoelectric effects or inherent nonlinearities enables the use of these systems across a diverse array of materials, expanding their potential applications well beyond the realm of quantum technologies. Another electro-optomechanical photon transducer has also been demonstrated on an SOI platform with a conversion efficiency of $1.72\;\pm \;0.14\times 10^{-7}$ in a 3.3 MHz bandwidth at room temperature [31]. In this system (Figure 4h), microwave photons resonate to stimulate a phononic crystal oscillator via electrostatic force facilitated by a charge-biased narrow-gap capacitor. The induced mechanical vibrations propagate through a phonon waveguide to an OMC, where they are subsequently converted into optical photons within the sideband of a pump laser field [31].

Compared to membrane, phononic crystal-based transducers often have larger bandwidths (~MHz), especially in the portion of the structure leveraging the piezoelectric effect. The primary advantage of phononic crystal resonators lies in their compatibility, which allows for support of various optomechanical coupling structures and diverse material selections, offering limitless possibilities for further system development [4,74,75]. Furthermore, transducers utilizing the piezoelectric effect usually exhibit lower noise than electro-optic ones and often do not require complex active cooling techniques [22,76,77,78].

However, in transducers using a “two-stage mode”, phononic waveguides inevitably introduce some losses. This separation also results in the microwave cavity lacking enhanced coupling structures, leading to relatively lower electromechanical synergy, and limiting the overall conversion efficiency of the transducer.

**Table 1 micromachines-15-00485-t001:** Comparison of OMC transducers.

Reference	Vainsencher et al. [69]	Jiang et al. [66]	Forsch et al. [36]	Jiang et al. [34]	Mirhosseini et al. [29]	Jiang et al. [36]	Weaver et al. [33]	Stockill et al. [67]	Arnold et al. [61]	Zhao et al. [31]
Platform	AIN	LN	GaAs	LN	AlN	LN	LN	GaP	SOI	SOI
$g_{om}/2\pi$ [MHz]	~0.115	0.12	1.3	0.08	0.5	0.41	0.53	~0.7	0.66	~ 0.577
$\kappa_{o}/2\pi$ [GHz]	15.2	0.78	5.8	1.2	0.8	1.12	~1.34	4.17	1.4	1.39
γ/2π [kHz]	5000	500	200	193	19	1100	~2630	~ 67	15	~ 0.001
$C_{om}$	$3\times 10^{-3}$	>1	1.73	$5\times 10^{-3}$	0.04	−	≪1	1.74	0.9	$2.7\times 10^{-4}$
$C_{em}$	$6\times 10^{-4}$	>1	−	−	~ 150	−	~24.2	−	0.57	$4.8\times 10^{-7}$
η	$10^{-8}$	$10^{-8}$	$<3.5\times 10^{-10}$	$10^{-5}$	$8.8\times 10^{-6}\;$*	0.05	0.09	$6.8\times 10^{-8}$	$1.9\times 10^{-4}$	$1.7\times 10^{-7}$
$\eta_{int}$	$6.5\times 10^{-3}$	$7\times 10^{-4}$	$7.2\times 10^{-2}$	$2.6\times 10^{-2}$	$10^{-3}$	0.35	−	−	$1.6\times 10^{-2}$	
$N_{add}$	$\sim 10^{4}$	−	−	−	0.57	~ 100	~7	~0.36	$1.9\times 10^{4}$ (M–O) $1.2\times 10^{4}$ (O–M)	−
T [K]	300	20	0.02	300	0.015	≤0.01	0.25	4	0.15	0.02

* Conversion from superconducting qubits to optical phonons; other data is about conversion from microwave photons.

### 2.3. Microdisk

Although less common in optomechanical transducers, microdisk structures are actually quite mature in their application in electro-optic transducers [9,79,80,81]. One major advantage of micro-disk resonators lies in their ability to operate at frequencies as high as 10 GHz (whereas membrane and OMC systems are always limited to a few gigahertz), implying lower cooling requirements and better compatibility with microwave frequencies [77,82].

Its mechanical resonator responds to electric field driving, although utilizing the piezoelectric effect, its operational mode closely resembles that of electro-optomechanical resonators [37]. By aligning the microdisk with the capacitor pad of a planar LC oscillator, which is constructed in a unique “ Ouroboros” structure (As shown in Figure 5b, the superconducting microwave resonator (yellow) forms a closed loop, connected end-to-end, resembling an ouroboros), cavity-enhanced electromechanical coupling is achieved, reaching a high electromechanical cooperativity of $C_{em}\;\sim \;7$ [37,83].

In optomechanical coupling, microdisk resonators are also referred to as whispering gallery mode (WGM) resonators. They utilize total internal reflection to confine incident light within a cavity with rotational symmetry. When the light interacts with the material inside the microdisk resonator, it exerts a small force, causing radial-breathing mechanical vibration in the microdisk [51]. These vibrations are coupled with the resonant modes of light, leading to energy conversion between light and mechanical vibrations, as shown in Figure 5a [84]. Microdisk resonators exhibit enhanced field density, which contributes to the enhancement of various nonlinear interactions [80].

Utilizing fundamental acoustic standing-wave modes in membrane thickness, AlN microdisks have achieved 10 GHz high-frequency mechanical resonances, as shown in Figure 5b. These resonances facilitate resonant coupling with photons simultaneously in a superconducting cavity and a nanophotonic cavity. The optical coupling quality factor and thickness mode mechanical quality factor are measured at $Q_{o,c}=5.8\times 10^{5}$ and $Q_{m}=1.1\times 10^{4}$, respectively [37]. An optical pulse pump was employed to increase the intracavity photon ($n_{cav}∼10^{5}$), enhancing the internal efficiency to approximately 5.8% and achieving an overall efficiency of $\eta \;\sim \;7.3\times 10^{-4}$ within a 1 MHz wide conversion bandwidth. The added noise is a few photons ($N_{add}=1.6$) at cryogenic temperatures (∼1 K), and the system’s optomechanical cooperativity ($C_{om}\;\sim \;0.4$) needs to be increased to effectively reduce mechanical noise during optical-to-microwave photon conversion [37]. The structure was also used in the optical phono-to-mechanical resonance of a spin qubits-to-telecommunication wavelength photons conversion at room temperature [38]. The minimized volume microdisk of diameter 5.3 μm maintains the low optical loss needed for coherent optomechanical interaction, with a higher phonon–photon coupling rate $g_{om}∕2\pi ∼\;$25 kHz and a mechanical radial breathing mode Q factor of $Q\;\sim \;10^{5}$ [38].

In conclusion, microdisk resonators demonstrate the potential for strong electromechanical coupling to convert microwaves to acoustic signals efficiently. Similar to FP cavities, the extended area of these membrane structures allows for effective coupling with superconducting resonators through the piezoelectric effect [38,85], though the larger mode volume in thickness mode-based resonators yields a reduced single-photon coupling rate ($g_{om}∕2\pi ∼$ 19 kHz) compared to OMCs [37].

### 2.4. Bulk Acoustic Wave Resonator (BAR)

BAR can mediate high-frequency elastic standing-wave acoustic modes (formed by the reflection of long-lived longitudinal acoustic modes between polished surfaces of a crystalline substrate) and standing-wave electromagnetic modes (formed within a FP optical cavity), with a high-frequency acoustic mode (8~13 GHz) [40,86,87]. The operation of BARs are based on the phenomenon of resonance; with the assistance of piezoelectricity, microwave fields can couple directly to the acoustic mode, enabling coherent control of phononic states and generation of phonon Fock states [39,88,89]. Furthermore, using Brillouin scattering, strong optical interactions with acoustic modes can be achieved [39,87,90].

Such a system has realized strong coupling between a single optical mode and one or more high-frequency (~GHz) modes of quartz BAR for bidirectional microwave–optical conversion at cryogenic temperatures [39,40]. Additionally, optical cavity enhancement of electromechanical synergy has been realized [39]. As depicted in Figure 6b, this structure has been used in the modular design of piezoelectric optomechanical systems, achieving both piezoelectric and Brillouin coupling in x-cut quartz, thus accomplishing bi-directional microwave–optical conversion with conversion efficiencies up to $10^{-8}$ at $C_{om}$ exceeding one [39]. Blesin et al.’s [91] recent work has demonstrated significant progress, achieving a bandwidth of 25 MHz and a conversion efficiency of $1.6\times 10^{-5}$ at room temperature. Moreover, they were able to generate nonclassical correlated microwave–optical photon pairs. Their structure utilizes low-loss silicon nitride ($S_{3}N_{4}$) photonic molecule, offering advantages such as compatibility with CMOS, compact size, and high on-chip power (>0.1 pW).

This structure has shown promising research potential, especially in terms of enhancing $C_{em}$. Utilizing different piezoelectric materials or optimizing the structure is expected to increase the existing $C_{em}$ by nearly two orders of magnitude (currently, $C_{em}\;\sim \;10^{-7}$) [39]. If the thinner optical waveguide is used to reduce acoustic scattering losses, both the mechanical Q factor and the microwave extraction efficiency can be increased [91,92]. Additionally, this system can consist of components composed of different acoustic, optical, and microwave resonators, rather than an integrated nanomechanical platform. This allows for independent fine-tuning of each resonator, including optimizing the Q factors and mode arrangement [39].

## 3. Discussion and Conclusions

Research and development in the field of quantum transducers has been thriving, especially in the past decade, when significant leaps were made, yielding numerous encouraging outcomes. Among various transduction schemes, the performance of optomechanical quantum transducers cannot be ignored. Of course, the ideal quantum transducer has yet to be realized, with conversion efficiency requiring improvement and the variety of quantum bits involved in coupling eagerly awaiting expansion (currently limited to superconducting qubits). The future development path of optomechanical quantum transducers warrants discussion.

Among four classes of optomechanical quantum transducers based on mechanical resonators, membrane resonators undoubtedly excel in conversion efficiency (see in Table 2). This structure was the first to achieve bidirectional microwave–optical conversion and has achieved conversion efficiency closest to the threshold (50%) [22,56]. Its high $C_{om}$ and cavity-enhanced electromechanical coupling are its greatest advantages. However, a lower bandwidth (kHz) and integration difficulties may limit future development. Additionally, this structure exhibits significant additional photon noise ($N_{add}\gg 1$), still far from the ideal sub-photon level, which may be addressed through better active cooling techniques.

Turning our attention to optomechanical crystals, we find that they can simultaneously localize optical and mechanical modes on the nm scale, possessing a significantly higher optomechanical coupling rate ($g_{om}/2\pi \;\sim \;$MHz) than other structures. This structure can support microwave-mechanical coupling in both electrical and piezoelectric approaches (membrane, microdisks, and bulk acoustic wave resonators only support single-mode electromechanical coupling), thus having great potential for improving electromechanical coupling rates. Another major advantage is its excellent material compatibility, conducive to large-scale integrated on-chip preparation [22]. However, electromechanical coupling using the piezoelectric effect faces high losses in the phonon waveguide. The separation of optomechanical and electromechanical coupling reduces noise but also reduces cooperativity. In conclusion, this structure should hold vast potential for development, and the achievement of microwave–photon entanglement has further propelled this structure to the forefront of the field. The next step in optimization may be to pursue higher Q factors and optimize coupling efficiency at each stage.

The optomechanical coupling rate, optomechanical cooperativity, and electromechanical cooperativity of microdisk resonators fall between those of membrane resonators and phononic crystals, while also exhibiting barely satisfactory conversion efficiency and bandwidth. This provides a new perspective for possible structures of optomechanical quantum transducers.

Transducers mediated by BAR exhibit structural flexibility and have several demonstrated advantages, including high-frequency acoustic mode, low cooling requirements, and high bandwidth [39,91]. Furthermore, by selecting different piezoelectric materials and optimizing the structure (such as using a thinner optical waveguide), significant improvements in parameter values may be achieved, rendering them worthy of further research [39,91].

When comparing the performance parameters of quantum transducers, their application direction is also a factor that needs to be taken into consideration. For instance, if the goal of a quantum transducer is precision measurement, then the efficiency of converting a single microwave photon to an optical photon becomes crucial [3]. Conversely, in integrated quantum networks, the bandwidth might need more attention.

In summary, various structures of optomechanical quantum transducers have achieved remarkable experimental progress. With researchers around the world continuing to delve deeper into this field, it is foreseeable that the realization of true bidirectional, direct microwave-to-optical (M–O) conversion is imminent, which will significantly advance our steps towards a large-scale “quantum Internet”.

## Figures and Tables

**Figure 1 micromachines-15-00485-f001:**
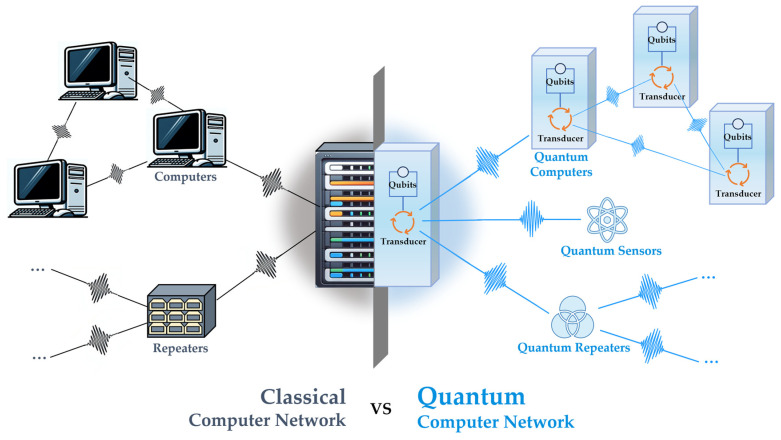
Comparison of classical and quantum computer networks. The left illustrates the classical computer network where computers connect and communicate with each other via fiber-optic network interface. The right shows a quantum computer network, where transducers are used to convert quantum information for communication between the quantum server and other devices such as quantum computers, quantum sensors, and the quantum repeaters that can work as nodes in the network.

**Figure 2 micromachines-15-00485-f002:**
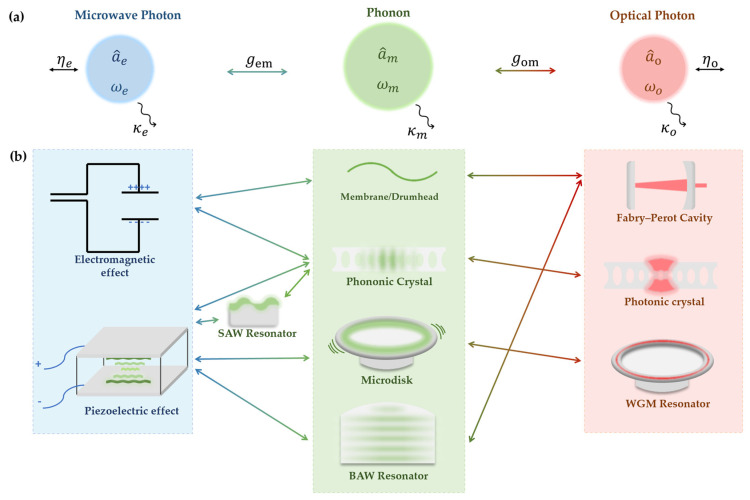
Schematic representation of the microwave-to-optical optomechanical transducer models. (**a**) Frequencies $\omega_{i}/2\pi$, bosonic operators ${\hat{a}}_{i}$, and loss rates $\kappa_{i}$ of three coupled modes of the transducer: microwave (blue, i=e), optical (red, i=o), and mechanical (green, i=m). The microwave and optical photons couple to external channels with efficiency $\eta_{e}$ and $\eta_{o}$, and the mechanical mode with coupling rate $g_{em}$ and $g_{om}$, respectively. (**b**) Different models (platforms) during microwave-mechanical-optical conversion are covered in this review. In the middle are the mechanical resonators capable of simultaneously coupling with microwaves and optics. Specifically, SAW resonators, which can only achieve electromechanical coupling, are listed separately.

**Figure 3 micromachines-15-00485-f003:**
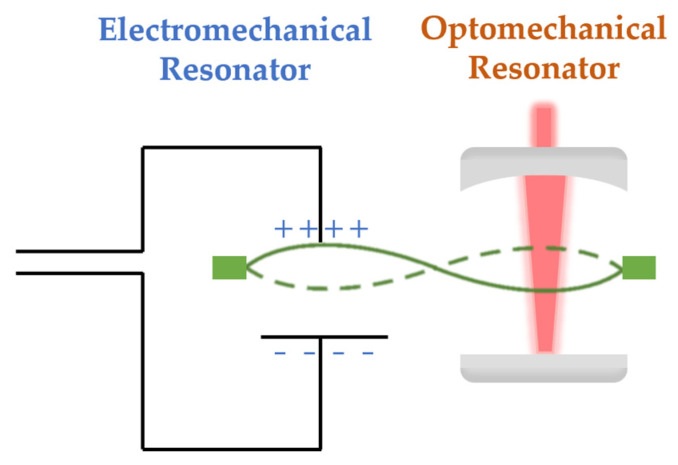
Membrane optomechanical systems: electromechanical resonator couples with FP optomechanical resonator using membrane as medium.

**Figure 5 micromachines-15-00485-f005:**
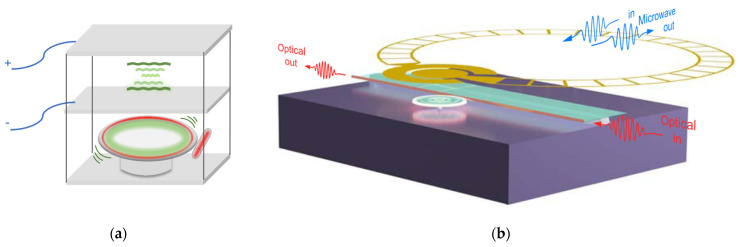
Microdisk cavity mediated optomechanical transducers. (**a**) Schematic diagram of microwave, acoustic, and optical modes of piezoelectric microdisk–WGM transducer. (**b**) AlN optomechanical microdisk cavity microwave–optical transducer. Reproduced from [37] under Creative Commons Attribution 4.0 International License.

**Figure 6 micromachines-15-00485-f006:**
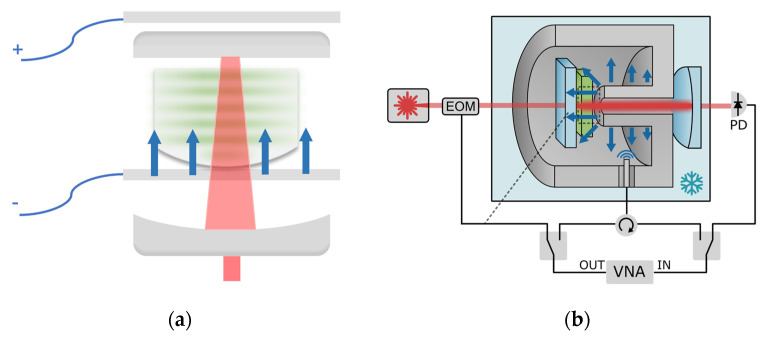
BAW resonator mediated optomechanical transducers. (**a**) Simplified experimental schematic of the hybrid piezo-Brillouin optomechanical system. (**b**) Schematic diagram of microwave, acoustic, and optical modes of piezoelectric BAW-FP transducer. Reproduced from [39] under the terms of the Optica Open Access Publishing Agreement.

**Table 2 micromachines-15-00485-t002:** Microwave–optics conversion parameters of different mechanical systems.

**Mechanical Resonator**	$\frac{g_{om}}{2\pi }\;$[kHz]	$C_{om}$	$C_{em}$	η	$\eta_{int}$	$N_{add}$	BW [MHz]	T [K]
Membrane [25,26,54]	0.06	132	132	$1.9\times 10^{-3}$ *	0.99	38	0.012	0.035
Phononic Crystal [29]	500	0.04	0.57	$8.8\times 10^{-6}$ *	$10^{-3}$	0.57	1	0.015
Microdisk [37]	19	0.4	7	$\sim 7.3\times 10^{-4}$	$\sim 5.8\times 10^{-2}$	1.6	~1	1
BAR [91]	0.42	$\sim 8\times 10^{-3}$	-	$\sim 1.6\times 10^{-5}$	$2\times 10^{-3}$	-	25	Room Temperature

* Conversion from superconducting qubits to optical photons; other data is about conversion from microwave photons.

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
