# Peer review of "Optomechanical Microwave-to-Optical Photon Transducer Chips: Empowering the Quantum Internet Revolution"

_micromachines, 2024, doi:10.3390/mi15040485_

Round 1
Reviewer 1 Report
Comments and Suggestions for Authors
The manuscript aims to give a brief overview of experimental advances in quantum transduction between microwaves and optical fields mediated by mechanical oscillators.
In my opinion, the manuscript does a reasonably good job of providing references to the most notable experimental results of recent years and highlighting the experimental groups that made important contributions to the field.
On the other hand, I think there are very many issues with the text. I compiled a (very likely incomplete) list which is at the end of the document ("Remarks/criticisms") and will try to outline the main structure of these issues in the numbered list below.
1. I think that the text lacks systematicity. The text is built around discussing (i) quantum transducers built using (ii) opto- (electro-) mechanical systems. The text has an optomechanical Hamiltonian [Eq.(2)] (electromechanical, while being essentially the same, is not mentioned), but the corresponding section ends very abruptly and the reader is left puzzled, what was the reason for writing this Hamiltonian. It is not mentioned anywhere that most transducers are designed to operate with red-detuned driving. The properties of a transducer are not introduced: what is the efficiency (internal, external), bandwidth, added noise, etc? Moreover, the properties of the transducer typically follow from its geometry (e.g. bandwidth is usually set by the effective linewidth of the mechanical oscillator). In the present manuscript, this conclusion is implied but never explicitly stated. In general, the manuscript is built around listing numerical properties of the most advanced published experiments, with little connection to the underlying physics.
2. The authors seem to use some unconventional terminology, e.g. 'thin film' optomechanics when speaking of membranes, others in the list below.
3. I think the writing style should be revised, made more scientific, and supported by references. At the beginning of each section, the authors write a few introductory sentences that are difficult to understand, and at least questionable (very possibly, even wrong; see remarks to lines 66, 96, 134, etc in the list below). These sentences strongly remind ones typically produced by AI language models. These sentences have to be rewritten and supported by references or proofs. The discussion and conclusion seem to contain a lot of the authors' opinions. E.g., in line 433 "BARs are relatively niche".
4. As a continuation of the first point, the authors' suggestions like in line 426, "the next step [...] may be to pursue higher Q factors" do not seem justified. While it is generally acknowledged that higher Q-factors are beneficial, it is not clear at all, why is this a priority, but not something else.
5. As a minor remark, I think two references are missing (see them at the very end).
To summarize, in the present form, the manuscript presents a reasonable compilation of references to important experiments published in recent years.
Tables 1 and 2 could be useful if the quantities in them were defined (there does not seem to be a single statement in the whole manuscript regarding what is $\kappa$ and what is $\gamma$.)
I do not see an explanation of the physics of the transducers' operation (beyond trivial statements that mechanics couples to both light and microwaves).
## Remarks/criticisms
- On p.2 the first sentence of the second paragraph 'However, the fragile ...' is questionable.
- I do not think Fig.1 is informative, though this is opinionated of course
- "utilizing principles such as optomechanics ..." sounds strange, optomechanics is not really "a principle"; perhaps "principles" → "platforms"?
- line 66: "Due to relatively long coherence time [...] this system can achieve efficient information transfer" --- this statement needs a proof. Long coherence time itself only allows [approximately coherent] operation of mechanical oscillators at timescales that are rather slow compared with optics or microwaves, but it is difficult to see how the length of coherence times contributes to the efficiency of the transfer. There are many factors that contribute, one of the main being the ability of mechanical oscillators to couple equivalently well to the fields of different wavelengths, and I disagree with the authors promoting long coherence times as the main role.
- Ref. 41 cites the review of a textbook by Bowen \& Milburn written by Shore, while the authors likely meant to make a reference to the textbook itself.
- line 96: "Phonons have gained favor in the quantum realm in recent years due to advantages such as being easily localized, prone to interaction, and capable of strong coupling with multiple physical dimensions" --- it is strange to read that "phonons have gained favor", and it is not clear what are multiple physical dimensions.
- why introduce minus twice, in Eq.(1) and in the definition of $H_{int}$?
- line 119: "When there is the cavity is driven"
- line 121: "magnetic field"; "driving $\Delta$" → "detuning $\Delta$"
- line 122: "diverse interactions": which interactions and what should be $\Delta$?
- line 124: linearization is around mean classical displacements which need not be the steady state
- line 129. In the community, the conventional term is 'membrane-in-the-middle' optomechanics, I am not sure that calling it a thin-film resonator is a good idea. Thin film production seems to be focused on the optical properties of the films, almost never on the mechanical oscillator properties which is a key requirement for optomechanics.
- line 134. "These vibrations can be longitudinal, transverse, or torsional, depending on the fixation method of the thin film, the type of external force, and the physical properties of the thin film itself"
* what is the use of torsional vibrations for opto-microwave transduction?
* what is the external force that the authors mean?
- line 143. "mechanical resonations" → phonons
- line 153. "Andrew" → "Andrews"
- line 156. It is incorrect to write "their work" here.
- line 158. Quantities $\eta_{o,e}$ are undefined at this point
- line 161-170. The paragraph should be clarified. Why "isolation" is in quotation marks? What does this sentence mean in the first place? What are ring resonators?
- line 185. Where does the number 99\% come from?
- line 198. "Phononic crystals are currently among the most favored mechanical resonators by researchers, which is composed of periodically arranged atoms or molecules." Periodically arranged atoms have nothing to do with photonic crystals. Phononic crystals typically have periodic patterns of artificial defects that result in bandgaps.
- line 222. The work by Bochmann (2013) seems earlier than Balram (2016)
- line 238. The cooperativity is not defined.
- line 285. 'yet-to-be-published' is likely going to become obsolete very soon. The sentence that starts on this line seems to be grammatically incorrect.
- line 310. "lower noise" than what?
- line 311. Ref 75 is purely optomechanics (not a transducer)
- line 312. What does "two-stage" mean?
- line 320. Could the authors elaborate what are the "acoustic thickness" modes?
- line 325. The authors may want to explain what is the "Ouroboros" structure.
- line 342. '105' should be $10^5$.
- Fig.5 is difficult to read: small font size and low image quality
- line 363. All of the systems in this review are high-Q resonators, why mention resonances only for BAW resonators?
- line 396. 'Less than a decade' seems to be derived from subtracting the publication year of Balram (2016) from 2024, which apparently misses the hard systematic work that started in the domain much earlier (works by Schwab, Roukes, Cleland to name a few)
- line 398. "Undoubted forerunners" is questionable.
- line 404. "form microwave" → "from microwave"; "phonons" → "photons"
- line 405. "Among four classifications" → "among four classes"
------
## Missing references
1. Bochmann, J., Vainsencher, A., Awschalom, D. D. & Cleland, A. N. Nanomechanical coupling between microwave and optical photons. Nat Phys 9, 712--716 (2013).
2. Blésin, T. et al. Bidirectional microwave-optical transduction based on integration of high-overtone bulk acoustic resonators and photonic circuits. Preprint at https://doi.org/10.48550/arXiv.2308.02706 (2023).
Try using more conventional terminology.
Author Response
Dear Reviewer,
Thank you for careful reading of our manuscript and for the positive comments on the significance and the quality of our work. We have revised the manuscript carefully according to the suggestions. All changes are attached to our reply letters, and are also applied in the revised manuscript. Hopefully our revised manuscript properly replies to the concerns of our referees, and satisfies the requirement for the publication in Micromachines.
Once again, we truly value your careful evaluation of our work and look forward to any further insights or comments you may have.
Sincerely,
Dr. Guangwei Deng on behalf of all authors.
Lab of Quantum Physics and Engineering, University of Electronic Science and Technology of China
March 25, 2024

Reviewer 2 Report
Comments and Suggestions for Authors
This is a review paper on optomechanical microwave-to-optical photon transducer. Recently, because of the rapid development of quantum computing devices, people in the quantum information science community have been interested in quantum networks. Since superconducting qubits only directly couple microwave photons, microwave-to-optical photon transducers are essential to building quantum networks. In this sense, the topic of this review paper is timely. Although the paper does not have so many pages, it gives most of the recent experiments of the field and discusses the differences among them. Further, the paper is written very well, and I did not find a part which should be updated. Hence, this paper will contribute the community, and I recommend the acceptance of this paper.
Author Response
Dear Reviewer,
Thank you very much for your positive and encouraging feedback on our work and paper. We truly appreciate your recognition of the timeliness and importance of the topic we addressed in our review paper. We wholeheartedly agree with your perspective that quantum transducers play a pivotal role in the development of quantum networks, especially given the increasing interest in quantum computing devices within the quantum information science community.
Furthermore, we would like to inform you that based on the suggestions provided by another reviewer, we have made certain revisions to the original manuscript. These changes aim to enhance the clarity and systematicity of the paper. If you are interested, you can download the latest version of the manuscript from the official website and read it.
Once again, we sincerely appreciate your thoughtful evaluation of our work, and we look forward to any additional feedback or comments you may have.
Best regards,
Dr. Guangwei Deng on behalf of all authors.
Lab of Quantum Physics and Engineering, University of Electronic Science and Technology of China
March 25, 2024
Round 2
Reviewer 1 Report
Comments and Suggestions for Authors
I find the text improved, however, insufficiently.
A clear unambiguous definition of the figure of merit of a transducer is still missing. In particular, the authors do not define efficiency of a transducer, nor its bandwidth.
The authors may be interested in consulting [1] for inspiration
[1] Zeuthen, E., Schliesser, A., Sørensen, A. S. & Taylor, J. M. Figures of merit for quantum transducers. Quantum Sci. Technol. 5, 034009 (2020).
The newly introduced text requires proof-reading, e.g.
- l. 99 "easy localization, prone to interaction" is likely grammatically incorrect
- there must be summation over j in Eq. (1)
- l. 125 should be "frame rotating with the drive"
- l. 126 should be "linearized Hamiltonian becomes" (this is the Hamiltonian of not only field, but field + mechanics, and not only optical [can be microwave])
- l. 145 pluralization "entanglements" is strange
- l. 146 "zero detuning" is confusing with $\Delta = 0$.
- Eq. (5) --- where does it come from? How is it physically defined?
- l.156--160: How is bandwidth defined (not determined)? What does it mean
- l. 170 "resonate" → "resonance"
- l. 184 why "breathing" here?
- l. 230 I still do not understand why "isolation" is in quotation marks
- l. 294 "considered low" --- they are not
See suggestions in the review.
Author Response
Dear Reviewer,
We are thankful for your thorough examination of our manuscript and your affirming remarks on our work. Gratefully acknowledging the references you provided, we have once again meticulously revised our manuscript in accordance with the suggestions presented. We have outlined all modifications in our response letters, and they have been duly integrated into the revised manuscript. We aspire that our adjustments aptly address the referees' queries and meet Micromachines' publication standards. Our sincere thanks once more.
Sincerely,
Dr. Guangwei Deng on behalf of all authors. Information and Quantum Laboratories, University of Electronic Science and Technology of China
March 28, 2024
